# The Androgen Regulated lncRNA *NAALADL2-AS2* Promotes Tumor Cell Survival in Prostate Cancer

**DOI:** 10.3390/ncrna8060081

**Published:** 2022-12-01

**Authors:** Levi Groen, Viktor Yurevych, Harshitha Ramu, Johnny Chen, Lianne Steenge, Sabrina Boer, Renske Kuiper, Frank P. Smit, Gerald W. Verhaegh, Niven Mehra, Jack A. Schalken

**Affiliations:** 1Department of Urology, Radboud Institute for Molecular Life Sciences, Radboud University Medical Center, P.O. Box 9101, 6500 HB Nijmegen, The Netherlands; 2Department of Genetics, Radboud University Medical Center, P.O. Box 9101, 6500 HB Nijmegen, The Netherlands; 3MDxHealth B.V., 6534 AT Nijmegen, The Netherlands; 4Department of Medical Oncology, Radboud University Medical Center, P.O. Box 9101, 6500 HB Nijmegen, The Netherlands

**Keywords:** prostate cancer, castration resistance, long noncoding RNA, apoptosis, survival, transcriptomics

## Abstract

Castration resistance is the leading cause of death in men with prostate cancer. Recent studies indicate long noncoding RNAs (lncRNAs) to be important drivers of therapy resistance. The aim of this study was to identify differentially expressed lncRNAs in castration-resistant prostate cancer (CRPC) and to functionally characterize them in vitro. Tumor-derived RNA-sequencing data were used to quantify and compare the expression of 11,469 lncRNAs in benign, primary prostate cancer, and CRPC samples. CRPC-associated lncRNAs were selected for semi-quantitative PCR validation on 68 surgical tumor specimens. In vitro functional studies were performed by antisense-oligonucleotide-mediated lncRNA knockdown in hormone-sensitive prostate cancer (HSPC) and CRPC cell line models. Subsequently, cell proliferation, apoptosis, cell cycle, transcriptome and pathway analyses were performed using the appropriate assays. Transcriptome analysis of a prostate cancer tumor specimens unveiled *NAALADL2-AS2* as a novel CRPC-upregulated lncRNA. The expression of *NAALADL2-AS2* was found to be particularly high in HSPC in vitro models and to increase under androgen deprived conditions. *NAALADL2-AS2* knockdown decreased cell viability and increased caspase activity and apoptotic cells. Cellular fractionization and RNA fluorescent in situ hybridization identified *NAALADL2-AS2* as a nuclear transcript. Transcriptome and pathway analyses revealed that *NAALADL2-AS2* modulates the expression of genes involved with cell cycle control and glycogen metabolism. We hypothesize that the nuclear lncRNA, *NAALADL2-AS2,* functions as a pro-survival signal in prostate cancer cells under pressure of targeted hormone therapy.

## 1. Introduction

Prostate cancer (PCa) constituted 27% of new cancer cases in US men throughout 2021, ranking at number one in this cohort [1]. The treatment backbone for advanced PCa is androgen deprivation therapy (ADT). Androgen deprivation reduces serum androgens to castrate levels, thereby inhibiting androgen receptor (AR) signaling and PCa proliferation. While initially successful, up to 20% of patients become resistant to ADT and progress to so-called, metastatic CRPC within 5 years of treatment [2]. Metastatic CRPC is characterized by substantial morbidity, with a median survival of only 24–36 months despite registration of multiple second-line therapeutic approaches [3].

Despite prolonged exposure to castrate conditions, PCa tumors continue to rely on AR signaling [4]. Gene amplification, expression of AR splice variants, de novo steroidogenesis and non-canonical signaling can all sustain AR signaling, promoting survival under castrate conditions via transcriptional control of proliferation and cell death [5,6]. Modulation of the AR transcriptional program by noncoding RNAs has also been shown to confer resistance to ADT [7,8]. In particular, lncRNAs have been found to support AR transcription, drive therapy resistance, and promote proliferation in PCa [7,9,10].

For example, the lncRNA *CCAT1* functions as a scaffold for the AR-transcription factor complex and promotes the expression of AR target genes under castrate conditions [9]. The expression of androgen receptor splice variant 7 (ARv7), a major driver of resistance to AR-targeted therapy, is dependent on the presence of lncRNA *MALAT1* [7,11]. Additionally, the lncRNA *PCA3* promotes PCa cell proliferation by suppressing the expression of tumor-suppressor p53 via the control of chromatin organization [10]. Together, lncRNAs support PCa progression under ADT through numerous molecular mechanisms. Investigation of novel lncRNAs upregulated under castrate conditions could yield new avenues for the development of future therapies and biomarkers.

In this study, an RNA-sequencing dataset containing CRPC, HSPC and benign prostate hyperplasia (BPH) samples was analyzed to identify lncRNAs upregulated in CRPC. Following semi-quantitative PCR (qPCR) validation with an expanded cohort, the lncRNA *NAALADL2-AS2* (hereafter named *AS2*) was selected for in-depth functional characterization. Tissue-specific expression of *AS2* was studied using The Cancer Genome Atlas (TCGA) and the Genotype-Tissue Expression project (GTEx) databases. Various hormone modulation and RNA interference models were used to identify drivers of *AS2* expression. The role of *AS2* on survival and apoptosis was assessed via *AS2* knockdown models. Quantitation of *AS2* transcripts in nuclear and cytoplasmic cell fractions, and RNA fluorescent in situ hybridization (RNA-FISH) were used to localize *AS2* in the cell. Finally, transcriptome and pathway analyses of *AS2* knockdown models were used to identify *AS2* target genes and pathways.

## 2. Results

### 2.1. AS2 Is Overexpressed in CRPC

Differential gene expression analysis using RNA sequencing data from CRPC (*n* = 4), high grade hormone-sensitive HSPC (*n* = 4), and benign/normal (*n* = 7) tumor RNA specimens led to the identification of several CRPC-associated lncRNAs (Figure 1 and Appendix A). *NAALADL2-AS2* (*AS2*) was the lncRNA that was most upregulated in CRPC compared to primary PCa (Figure 1A). To further characterize the context in which *AS2* is expressed, the Genotype-Tissue Expression (GTEx) and TCGA databases were analyzed [12]. Expression of *AS2* is increased in esophageal carcinoma, diffuse large B-cell lymphoma, lung squamous cell carcinoma, stomach adenocarcinoma, ovarian serous cystadenocarcinoma, and prostate adenocarcinoma. In contrast, *AS2* is not expressed in their normal tissue counterparts (Figure 1B). In a semi-quantitative PCR validation analysis using tumor-derived RNA, *AS2* was found to be significantly upregulated in CRPC when compared to normal, benign hyperplasia, low grade, and high grade HSPC (Figure 1C). Utilizing the prostate adenocarcinoma (PRAD) TCGA dataset, HSPC patients with high tumoral *AS2* expression have a shorter time to progression than those with low *AS2* expression (Figure 1D).

#### 2.1.1. AS2 Expression in PCa Cell Lines

To quantify the expression of *AS2* in the context of HSPC and CRPC, two cell lines mimicking HSPC (LAPC-4 and LNCaP), and six cell lines mimicking CRPC (22Rv1, DU145, DuCaP, MDA-PCa 2b, and PC3) were used in a quantitative gene expression analysis [13]. All cells were grown under normal conditions, i.e., in medium supplemented with fetal bovine serum (FCS). The expression of *AS2* was found to be relatively high in LAPC-4 and LNCaP cells, and low in the other cell lines, suggesting that *AS2* may be primarily important for HSPC cells (Figure 2). Based on these data, LAPC-4 and LNCaP cells are the most suitable HSPC models for further experiments using *AS2* knockdown, and DuCaP is the most suitable CRPC model for further analysis.

#### 2.1.2. AS2 Is Co-Expressed with NAALADL2 

*AS2* is located on the long arm of chromosome 3, in the antisense orientation of the oncogene *NAALADL2* (Figure 3A). To investigate whether these two transcripts are co-expressed, a correlation analysis was performed on gene expression data from 44 patient tumor-derived RNA specimens. There is a strong positive correlation between the expression of *AS2* and *NAALADL2* (*R* = 0.8, *p* < 2.2 × 10^−16^) (Spearman) (Figure 3B). The outliers in this analysis correspond to benign or metastatic samples. The low-grade, high-grade and CRPC samples generally fall along the regression line. As previous studies have identified many antisense lncRNAs to regulate their sense-gene neighbors, *NAALADL2* expression was analyzed following knockdown of *AS2* using GapmeR antisense oligonucleotides [14]. Interestingly, *AS2* does not appear to regulate the expression of *NAALADL2* (Figure 3C,D).

#### 2.1.3. Copy Number Gains around the AS2 Locus Are Highly Prevalent in PCa

Over 50% of all PCa patients have somatic alterations on the 3q26.31-32 locus, where the gene for *AS2* resides [15]. Spanning this locus, increased copy numbers (CN(s)) of *NAALADL2* and *TBL1XR1* have been identified in 20–30% of all CRPC and neuroendocrine prostate cancer (NEPC) tumors. Furthermore, tumors with increased CNs of this locus also show an increased expression of *NAALADL2* and *TBL1XR1.* Interestingly, of the six characterized cell lines, PC3 and DU145 have the highest CNs of this locus but are low in *AS2* expression (Figure 2 and Appendix A).

#### 2.1.4. Androgens Regulate the Expression of AS2

To investigate the transcriptional regulators of *AS2* expression, an in vitro model of ADT was designed. Androgen-sensitive LNCaP cells were grown in an androgen-depleted environment and then cells were stimulated with synthetic AR-ligand R1881. Androgen deprivation led to an increased expression of *AR* and a significantly decreased expression of *KLK3*, a canonical AR target gene (Figure 4A,B). *AS2* expression was significantly increased upon androgen depletion (Figure 4C). Similarly, LAPC-4 cells exposed to decreasing levels of R1881 increased their expression of *AS2* while decreasing *KLK3* expression (Figure 4D). Thus, under acute ADT, these hormone sensitive lines increase their expression of *AS2.* Curiously, castration resistant lines, LAPC4-CR and LNCaP-CR, show a decreased expression of *AS2* when compared to their hormone sensitive parental lines (Appendix A) [16].

#### 2.1.5. AR Does Not Directly Regulate AS2 Expression

Following androgen modulation experiments, the involvement of AR-mediated transcriptional regulation of *AS2* was investigated. The subsequent experiments were done using LNCaP and LAPC-4 as hormone-sensitive in vitro models. Using small interfering RNAs (siRNA) targeting *AR*, a significant 70% knockdown was achieved in both cell lines (Figure 5A,D). As expected, the expression of the AR target gene *KLK3* decreased following *AR* knockdown (Figure 5B,E). Unlike androgen deprivation, *AR* knockdown did not directly downregulate *AS2* expression (Figure 5C,F). The AR transcriptional regulatory complex consists of various co-factors such as FOXA1 and GATA2. Interestingly, these co-factors also have enhancer binding sites at the *AS2* transcriptional start site (TSS) (Appendix A). Hence, the transcriptional regulation of *AS2* by AR co-factors FOXA1 and GATA2 was investigated. Knockdown of *GATA2* using specific siRNAs leads to an approximate 50% reduction in *GATA2* expression in both LNCaP and LAPC-4 cells (Figure 5G,I). However, no significant effect on *AS2* expression was observed in *GATA2* knockdown cells (Figure 5H,J). Similarly, knockdown of *FOXA1* did not affect *AS2* expression in LAPC-4 or DuCaP cells (Appendix A).

#### 2.1.6. AS2 Promotes Survival by Inhibition of Apoptosis

To investigate the biological function of *AS2* in PCa, the androgen sensitive LAPC-4 and CRPC-like DuCaP cells were used. A 70% knockdown of *AS2* expression in LAPC-4 cells was achieved via GapmeR transfection targeting *AS2* RNA (Figure 6A). Due to the persistent negative impact on cell viability, *AS2* expression could not be measured following transfection in DuCaP cells.

The imbalance between cell proliferation and cell death is a major hallmark of all cancers [17]. In the present study, cell viability was used to assess the capacity for proliferation by quantitating cellular adenosine triphosphate (ATP) levels, and caspase -3 and -7 activity and DNA fragmentation were assessed as a surrogate for apoptosis.

Following *AS2* knockdown, cell viability decreases noticeably at day 3 post-transfection and reduces significantly at day 6 when compared to controls (Figure 6B). Caspase activity was increased two days after *AS2* knockdown in both LAPC-4 and DuCaP cells when compared to controls (Figure 6C,D). In line with these findings, *AS2* knockdown also significantly increased the percentage of cells with fragmented DNA, as deduced from the sub-G1 fractions in cell cycle profiling (Figure 6E). The percent of subG1 gated cells can be found in Appendix A, including the cell cycle histograms. The latter parameter could not be assessed in DuCaP cells due to excessive cellular debris following *AS2* knockdown.

#### 2.1.7. AS2 Is a Nuclear RNA

The cellular localization of a lncRNA can give a clue to the type of regulatory function it performs. To determine the localization of *AS2*, two complementary experiments were carried out; *AS2* expression was compared in isolated cytoplasmic and nuclear RNA fractions; and RNA fluorescent in situ hybridization (RNA-FISH) was used to image *AS2* transcripts in formaldehyde-fixed cell preparations. Cytoplasmic and nuclear RNA was isolated from LNCaP and MDA-PCa 2b cells for qPCR analysis. The nuclear RNA *RNU6* and the cytoplasmic RNA *GAPDH* were used as controls. As expected, *RNU6* was measured predominantly in the nuclear fractions. *GAPDH* mRNA was primarily found in the cytoplasmic fraction of LNCaP cells, but in both cytoplasmic and nuclear fractions of the MDA-PCa 2b preparation, the latter suggests cytoplasmic RNA contaminated of the nuclear fraction. Nevertheless, *AS2* transcripts were measured almost exclusively in the nuclear fractions in both cell lines (Figure 7A,B).

A combined RNA-FISH and immunofluorescence assay was designed to image *AS2* transcripts and cytokeratin -8 -18 -19 (CK) proteins in LAPC-4 cells. The low *AS2* expressing cell line PC3 was used as a negative control. Fluorescent probes targeting *AS2* are shown in green, fluorescent antibodies targeting CK are shown in red, and the nucleus stained by DAPI is shown in blue. As shown in Figure 7C, *AS2* probes localize into distinct foci in LAPC-4 nuclei. No *AS2* signal was observed in PC3 cells (Figure 7D).

#### 2.1.8. AS2 Is a Transcriptional Modulator

Nuclear lncRNAs have distinct regulatory functions. These functions can act upstream, downstream, or be directly involved with active transcription [18]. To investigate the regulatory role of *AS2* in the context of PCa, transcriptome analysis of *AS2* GapmeR-transfected LAPC-4 cells was performed. Total RNA from three independent transfections, 72 h post-transfection, were used. *AS2* knockdown was confirmed by qPCR analysis (data not shown). Due to potential off-target effects by the scrambled control oligo (SCR), this control was omitted from transcriptome analysis. The DESeq2 package was used to identify differentially expressed genes (DEGs), as described [19]. The adjusted *p*-value cutoff used to define DEGs was *p* < 0.001. There were 333 DEGs in *AS2* knockdown cells compared to the control group (Figure 8A). Most DEGs were upregulated following *AS2* knockdown. These findings were validated by qPCR analysis of *AS2* knockdown cells compared to scrambled control (Appendix A). Pathway analysis using the Reactome Pathway Browser identified TP53, CDKN1A (p21), and glycogen metabolism-related pathways to be deregulated following *AS2* knockdown (Figure 8B) [20].

## 3. Discussion

Castration resistance is the leading cause of PCa-related death. Sustained androgen signaling under ADT is a hallmark in the progression from HSPC to CRPC. Low levels of remaining androgens are likely the primary drivers of progression under ADT [21]. In fact, tumor dihydrotestosterone (DHT) levels have been shown to remain at 25% of pre-ADT levels 6 months into treatment [21]. In an effort to tackle this issue, ADT in combination with CYP17A inhibitors or AR antagonists are now standard of care in most western countries. Although modestly successful, there is room for improvement [22]. By directing research efforts towards unraveling the survival mechanisms employed by primary PCa cells under ADT, future lives may be spared. Clinical trials have largely mobilized around more potent AR antagonists, inhibitors of de novo steroidogenesis and more recently targeting AR splice variants as novel AR targeted therapies (NCT03123978, NCT02807805, NCT02566772, NCT03888612). Unfortunately, less than 1% of primary PCa patients are ARv7 positive. This number increases marginally to 3% of first line CRPC patients [23]. However, approximately 20% of all PCa patients will develop castration resistance within 5 years of their diagnosis [24].

It has been shown recently that noncoding RNAs (ncRNAs) constitute a major part of a cell’s transcriptional regulation. In the nucleus, ncRNAs can modulate transcription by scaffolding with other RNAs and proteins to guide or block access to chromatin, interact with active transcriptional sites, alter splicing, or stabilize messenger RNA (mRNA) [18]. For example, the lncRNA *PCGEM1* has been found to aid AR binding to the promoter regions of AR-regulated genes under ADT [25], while lncRNA *PCA3* suppresses the function of the tumor suppressor p53, of which the inactivation is central to PCa tumor progression [10,26]. Another example is *MALAT1*, which alters splicing to produce therapy resistant splice variant ARv7 [7]. Noncoding RNAs can form an integral part of a cancer cell’s adaptive response when exposed to a threat like ADT.

In this study, the lncRNA *AS2* was found to be upregulated in CRPC tumor biopsies when compared to low and high-grade PCa and benign prostatic tissue. In silico analysis using TCGA data suggest that *AS2* may be overexpressed in a number of other malignancies. In HSPC, *AS2* likely functions as a driver for progression, as patients with high expression of *AS2* have a shorter time to progression when compared to patients with low *AS2* expression. A previous publication identified the *AS2* overlapping protein coding *NAALADL2* gene as an oncogene in PCa [27], although the function of *NAALADL2* remains elusive. We found that *NAALADL2* and its antisense transcript, *AS2*, are co-expressed. Being an antisense lncRNA, we hypothesized that *AS2* functions as an mRNA stabilizer of *NAALADL2* [14]. Yet, knockdown of *AS2* did not affect *NAALADL2* expression levels. Cell viability, however, was markedly decreased upon *AS2* knockdown, which was accompanied by the increased induction of apoptosis. It seems that *AS2* functions to promote tumor cell survival. Transcriptome analysis of LAPC-4 cells with *AS2* knockdown identified 333 DEGs, the majority of which were upregulated. Amongst those upregulated DEGs, *CDKN1A, GZMM*, and *FOSB* are known modulators of apoptosis. In cancer cell lines, overexpressed *CDKN1A* has been found to enhance cell death responses following DNA-damage induced by cisplatin [28,29]. Furthermore, *CDKN1A* is transcriptionally controlled by tumor suppressor TP53, which controls key biological pathways related to cell cycle control and apoptosis. As shown in our pathway analysis, TP53-related pathways were significantly deregulated. Interestingly, amongst the few downregulated DEGs, *ZC3H12C* mapped to several TP53-related pathways. By modulating the expression of TP53-signaling genes, *AS2* could suppress apoptosis and promote the survival of PCa cells. Independent of TP53 signaling, *GZMM* and *FOSB* are known to promote apoptosis by caspase activation and as part of the AP-1 transcription factor complex, respectively [30,31]. Increased *FOSB* expression has been shown to be pro-apoptotic in MCF7 cells [32]. By modulating glycogen metabolism, *AS2* may also indirectly promote survival in PCa cells [33]. Amongst the *AS2*-modulated genes there were three enzymes involved with glycogen and carbohydrate metabolism, namely *PYGM, ENO,* and *LCT*. Glycogen is an important source of energy for cancer cells under stressful and hypoxic conditions [34]. In this light, ADT-induced *AS2* expression could preserve glycogen stores by the downregulation of glycolytic enzymes such as *PYGM*. In turn, increased glycogen storage may promote survival of ADT-stressed PCa cells. This hypothesis finds credence in a study from Schnier et al., who found that androgen deprivation led to glycogen accumulation in hormone-sensitive LNCaP cells, but not in hormone-insensitive PC3 cells [35]. In line with our findings, *AS2* expression increases in LNCaP cells exposed to ADT. Future studies could help elucidate the role of *AS2* in the context of ADT, glycogen homeostasis and PCa cell survival.

Over 50% of all PCa tumors bare an amplification in the 3q26.31-32 locus, a genetic region on chromosome 3 known for its high density of oncogenes including *NAALADL2* and *TBL1XR1.* Copy number (CN) gains of these genes are associated with their increased expression, consistent with the mechanism of self-regulating expression. *NAALADL2* is co-expressed with various AR-target genes while its genetic neighbor, *TBL1XR,* is a known AR co-activator [15]. This prevalent genetic alteration could also be a key driver of *AS2* expression. Indeed, we found *AS2* to be androgen regulated, but not via direct control of the AR or two of its major co-factors, GATA2 and FOXA1. *GATA2* remains to be a candidate of interest due to its association with *NAALADL2* expression and its binding motifs at the *NAALADL2* and *AS2* promoters. With this in mind, one could expect a positive association between CNs of *NAALADL2* and *TBL1XR1* and *AS2* expression. Yet, amongst 6 PCa cell lines characterized by the CCLE, DU145 and PC3 score highest on CNs for this region but are essentially negative for *AS2* expression. By focusing on TFs which are negatively regulated by androgens with binding sites near the *AS2* promoter, future studies may identify the drivers of *AS2*, and accordingly *NAALADL2*, expression. To summarize, we found *AS2* to be androgen regulated, promote tumor cell survival, and modulate the expression of genes involved with apoptosis and glycogen metabolism. Together, *AS2* could provide HSPC cells with an adaptive response to ADT by preventing apoptosis and promoting survival.

Due to remaining androgens under primary ADT and sustained AR signaling, androgen receptor signaling inhibitors (ARSIs) are used to enhance castration in first line CRPC therapy [36,37]. Successive lines of AR-targeted therapy induce an evolutionary selection for androgen-insensitive cells. In fact, clonal expansion of these cells has been shown to drive resistance to enzalutamide and abiraterone, two commonly used ARSIs [38]. Under the pressure of ARSIs, castration resistant cells gain an adaptive advantage. In this scenario, high *AS2* levels may help to identify patients with largely androgen-driven tumors and sensitivity to ARSIs. Vice versa, patients with low *AS2* levels might benefit from alternative treatments. In this light, castration resistant LAPC-4-CR and LNCaP-CR show decreased expression of *AS2* when compared to their parental lines, suggesting a redundant role for *AS2* in a castration resistant setting [16]. This could also explain the perceived discordance of *AS2* levels in CRPC tissue and CRPC-like cell lines. In the context of CRPC tissue samples, *AS2* expression is increased due to the exposure to first-line ADT. In contrast, CRPC-like cell lines are continuously cultured under hormone deprived conditions, akin to successive lines of AR-targeted therapy. Having adequately adapted to these conditions, *AS2* expression is likely to become redundant. As such, increased *AS2* expression constitutes part of an acute adaptive response for survival under ADT, a response which is not required for adequately adapted castration resistant cells. This hypothesis is supported by two of our recent publications. Here we found high plasma-*AS2* levels in CRPC patients treated with ARSIs to be favorable for progression free survival and overall survival, while low *AS2* levels were unfavorable [39,40]. These findings support the clinical utility of *AS2* as a liquid biopsy-based biomarker in CRPC. Future clinical trials should evaluate the potential of *AS2* as a liquid-biopsy based biomarker in the hormone sensitive setting.

## 4. Materials and Methods

### 4.1. Tumor lncRNA Sequencing Analysis

#### 4.1.1. Clinical Samples

The present study used archival biomaterials collected between 1980 and 2000 under an opt-out regulation at Radboudumc, Nijmegen, The Netherlands. All materials were anonymized. The clinical samples comprised a panel of samples from various stages of prostate disease. The panel included four non-malignant prostate samples (NP), three benign prostate hyperplasia (BPH) samples, four HSPC samples, four castration-resistant prostate cancers (CRPC) and four metastatic lesions (meta) samples. Upon radical prostatectomy or transurethral resection of tumor tissue, specimens were snap frozen in liquid nitrogen. Specimens were selected for purity of benign or cancer cells, respectively, and processed by step sectioning. Total RNA was isolated using Trizol reagent according to the manufacturer’s protocol (Invitrogen, Waltham, MA, USA).

#### 4.1.2. RNA Sequencing

Total RNA from clinical samples was subjected to DNase treatment with Turbo DNase to remove contaminating DNA. RNA was purified using an RNeasy MinElute Cleanup Kit (Qiagen, Hilden, Germany) and ribosomal RNA was removed using an Ribo-Zero Magnetic Gold kit (Westburg, Utrecht, The Netherlands). A 5× fragmentation buffer (200 mM Tris-acetate (pH 8.2), 500 mM potassium acetate and 150 mM magnesium acetate) was used to fragment RNA. First strand cDNA synthesis was performed using random hexamers, Superscript III and Actinomycin D, which was followed by sample clean-up and second strand cDNA synthesis involving dUNTPs. Next, an A-base was added to the 3′ end of the cDNA and Illumina adapters were ligated to the cDNA. The second strand was degraded using USER enzyme (Bioke, Leiden, The Netherlands) and a prePCR was performed to replace the second strand to ensure strand-specificity of the data. Size selection using the E-Gel iBase Power System was performed to obtain fragments of approximately 200 to 400 bp, depending on the sequencing protocol (Thermo Fisher Scientific, Waltham, MA, USA). Next, another PCR, using 10 to 12 cycles, depending on input quantity, was performed to enrich for fragments carrying adapters. Finally, the PCR product was purified using AMPure XP beads (Beckman-Coulter, Brea, CA, USA) and final concentrations were measured using Qubit fluorometric quantitation (Invitrogen, Waltham, MA, USA), while fragment lengths were determined by TapeStation (Agilent Genomics, Santa Clara, CA, USA). cDNA quality and rRNA depletion was ascertained via qPCR. If the samples passed this quality check, they were pooled at two samples per lane at a final concentration of 2 or 20 nM, depending on the sequencing protocol. Clinical samples were sequenced paired-end using 100 bp reads on the Illumina HiSeq 2000 at Maastricht UMC (Maastricht, The Netherlands).

#### 4.1.3. RNA Sequencing Data Analysis

Clinical samples sequenced in Maastricht returned on average 200 million reads. Reads were paired –end mapped to human genome hg19 as provided by the UCSC genome browser using the Genomic Short-read Nucleotide Alignment Program (GSNAP) [41].

For downstream analysis, genes were annotated using a modified version of GENCODE V17 for which all known isoforms of a gene (defined by ENSTs corresponding to the same ENSG) were combined to generate the longest possible transcript. RPKMs (read per kb per million) were calculated using the rpkmforgenes.py tool for each (adjusted) ENSG [42].

#### 4.1.4. Cell Culture

LNCaP (CRL-1740) and 22Rv1 (CRL-2505) cell lines were obtained from American Type Culture Collection (ATCC) (Manassas, Virginia, USA). DuCaP cells were kindly provided by Dr Ken Pienta (University of Michigan, MI, USA) and LAPC-4 cells by Dr Rob Reiter (University of California, CA, USA). All cells were cultured at 37 °C and 5% CO_2_ in a humidified atmosphere. Cell lines were authenticated using the PowerPlex 21 system (Promega) by Eurofins Genomics (Ebersberg, Germany). Cell lines were frequently tested for Mycoplasma infection, using a Mycoplasma-specific PCR, and cells were propagated for no more than 6 months or 30 passages after resuscitation from stocks. Details regarding culture media composition can be found in Appendix A.

#### 4.1.5. Androgen Modulation Assays

LNCaP and LAPC-4 cells were used for androgen modulation assays. Approximately 120,000 cells were plated per well in 12-well plates in 1 mL phenol red-free RPMI medium supplemented with androgen-containing (FCS) or androgen-depleted (CSS) serum and treated with synthetic AR ligand R1881 (Organon, Oss, The Netherlands) for 0 to 72 h.

#### 4.1.6. Androgen Taper Assays

LAPC-4 cells were cultured in 25 cm^2^ flasks in normal IMDM culture medium. With each subsequent passage, the concentration of R1881 in the culture medium was reduced from 1nM to 0nM of R1881.

#### 4.1.7. RNA Extraction

Fresh frozen tissue sections were cut into 20 micron sections for RNA extraction. Cultured cells were lysed in TRIzol reagent. RNA was extracted using TRIzol reagent as described by the manufacturer (Invitrogen, Waltham, MA, USA). Pelleted RNA was resuspended in RNAse-free milliQ water. RNA concentration and purity was determined using a NanoDrop1000 spectrophotometer (Thermo Fisher Scientific, Waltham, MA, USA).

#### 4.1.8. Reverse Transcription (RT) Reactions

Total RNA (0.5–2 ug) was first treated with DNaseI (Invitrogen) and then converted into single-stranded complementary DNA (cDNA) via random-primed reverse transcription, using Superscript II reverse transcriptase (Invitrogen). The produced cDNA was diluted 1:1 in RNAse-free milliQ water and stored at −20 °C. RNA samples not treated with RT were used as a control to test for genomic DNA contamination.

#### 4.1.9. Real-Time Quantitative Polymerase Chain Reaction (qPCR)

A real-time-PCR analysis was performed using LightCycler 480 SYBR Green I Master mix (Roche, Basel, Switzerland). Amplification and analysis were done on a LC480 LightCycler (Roche), using the following program: first 5 min at 95 °C, then 40 cycles of 10 s at 95 °C, 20 s at 60 °C and 10 s at 72 °C. All lncRNA and AR-related primers used are listed in Appendix A. Expression levels of *GAPDH* were used for normalization. Relative gene expression levels were calculated according to the mathematical model for relative quantification in real-time PCR described by Pfaff [43].

#### 4.1.10. Cytoplasmic and Nuclear RNA Isolation

Cytoplasm and nuclear RNAs were isolated as described previously [44].

#### 4.1.11. RNA Fluorescent In Situ Hybridization (RNA-FISH) and Immunofluorescence

Fluorescently labelled probes targeting *AS2* were used for RNA-FISH (Biosearch Technologies, Hoddesdon, United Kingdom) (Appendix A). Samples were prepared as described by the manufacturer’s protocol. Anti-cytokeratin (CK) -8 (ab192467), -18 (ab206091), and -19 (ab192643) antibodies were supplied by abcam and 0.66 μL was added directly to 100 μL of hybridization buffer. Each experiment included the following conditions: negative control (hybridization buffer), *AS2*, and CK8/18/19.

### 4.2. Transfection of GapmeR Antisense Oligonucleotides

Cell lines were transfected with 0.7 uM GapmeR antisense oligonucleotides (Eurogentec, Seraing, Belgium) using X-tremeGene9 DNA Transfection Reagent (Roche, Basel, Switzerland) following the manufacturer’s protocol. Sequence details can be found in Appendix A. Reagent control consisted of Opti-MEM and XtremeGene9 reagent.

#### 4.2.1. Transfection of siRNAs

Cells were transfected with 10nM small interfering RNAs (siRNAs) (Ambion, Waltham, MA, USA) using Lipofectamine RNAiMAX reagent (Invitrogen, Waltham, MA, USA) according to the manufacturer’s protocol. Sequence details can be found in Appendix A. Reagent control consisted of Opti-MEM and RNAiMAX reagent.

#### 4.2.2. Cell Viability Assay

To assess cell viability, 10,000 cells were cultured in 96-well culture plates. Transfection with oligonucleotides was done 24 h after cell seeding. Cell viability was measured at regular intervals using a CellTiter-Glo assay according to the manufacturer’s protocol (Promega, Madison, WI, USA). Luminescence was measured using a Victor3 multilabel reader (PerkinElmer, Waltham, MA, USA). To calculate the relative cell viability over time, luminescence readouts from day 0 were used as a reference. Each experiment was performed in triplicate and repeated at least three times.

#### 4.2.3. Apoptosis Assay

In parallel to the cell viability assays, cells were seeded into 96-well plates for assessment of Caspase-3/7 activity using the Apo-ONE Homogenous Caspase-3/7 Assay (Promega, Madison, WI, USA), following the manufacturer’s instructions. After 4 h of incubation, fluorescence was measured on a Victor3 multilabel reader (PerkinElmer, Waltham, MA, USA). The luminescence signal from medium alone was used for background normalization. Caspase-3/7 activity was normalized to values in scramble control-transfected cells. Each experiment was performed in triplicate and repeated at least three times.

#### 4.2.4. Cell Cycle Analysis

One day before transfection, 300,000 cells were seeded per well of a 6-well plate. The next day, cells were transfected with oligonucleotides as described above. Samples were harvested 5 days after transfection. Briefly, cells were washed with 0.9% NaCl and harvested using trypsinization. The medium was saved to keep apoptotic cells. Following centrifugation at 200× *g* for 5 min, cell pellets were resuspended in Hank’s balanced salt solution (Invitrogen, Waltham, MA, USA), and cells were fixated with ice-cold ethanol (58%). Fixated cells were centrifuged and washed in 500 μL PBS with 1% BSA and 2 mM EDTA. Washed cells were centrifuged for 5 min at 200× *g* and resuspended in 500 μL PBS with 1% BSA and treated with RNase A (100 μg/mL, Sigma, St. Louis, MO, USA) for 40 min at 37 °C. Subsequently, cells were stained with propidium iodide (40 μg/mL, Sigma, St. Louis, MO, USA) for 15 min in the dark. The samples were analyzed on a CytoFlex (13-color) flow cytometer (Beckman-Coulter, Brea, CA, USA), and histograms were created and analyzed using Kaluza Flow Analysis software (version 2.1.3) (Beckman-Coulter, Brea, CA, USA). Results were reproduced in four independent experiments.

#### 4.2.5. In Silico Analysis

Tissue-specific expression of *AS2* was studied using The Cancer Genome Atlas (TCGA) and the Genotype-Tissue Expression project (GTEx) databases using the GEPIA2 portal [12]. The Kaplan-Meier plot was created using GEPIA2 [12]. The bodymap was stylized using BioRender.com. Copy number alterations in cell lines were downloaded from the Broad Institute’s depmap portal, accessed on 2 May 2022 (https://depmap.org/portal/ccle/). Significantly differentially expressed genes identified via DESeq2 were used as input for the Reactome Pathway Browser to identify biological pathways regulated by *AS2* [20]. Pathways were reported based on overlapping results between the up versus down analyses as well as statistical significance and effect size of the DEGs. Figure 1, Figure 2, Figure 3 and Figure 8 were stylized with BioRender.com.

#### 4.2.6. Transcriptome Analysis

PolyA-enriched total RNA was used for RNA sequencing and differential gene expression analysis. Short read data sets were obtained using Illumina NovaSeq 6000 next generation sequencing and the RNA-Seq v4 pipeline that was developed in-house by GenomeScan B.V. (Leiden, The Netherlands). Read counts were loaded into the DESeq2 package to find differentially expressed genes between two conditions, *AS2* knockdown versus reagent control (*n* = 3 independent samples each).

#### 4.2.7. Statistical Analysis

Statistical analysis was used using GraphPad Prism 9 and Rstudio (version 2022.07.2 + 576). When not specified in the figure, statistical tests were done using a paired student’s *T* test in GraphPad Prism9. The violin plot and scatterplot were made using the ggpubr (version 0.4.0) package for RStudio. The volcanoplot was produced using the EnhancedVolcano package for RStudio (Bioconductor version: Release 3.16) [45].

## 5. Conclusions

The long noncoding RNA *AS2* is overexpressed in various cancers including PCa. In PCa, *AS2* expression is increased in high-grade and CRPC tumor specimens when compared to low-grade and benign prostate samples. In the HSPC setting, high *AS2* expressors have a shorter time to progression than low expressors. In PCa cell lines models, *AS2* is expressed at the highest levels in androgen-sensitive cell lines, with hormone deprivation leading to a further increase in *AS2* expression. Knockdown of *AS2* resulted in reduced cell viability and increased apoptosis. *AS2* is localized in the nucleus, where it is involved in the transcriptional regulation of various genes related to cell cycle control and glycogen metabolism. *AS2* likely functions to promote the survival of androgen sensitive PCa cells by preventing caspase activation and cell death, and therefore is a poor prognosticator for PCa patients. The molecular mechanisms driving *AS2* expression remain elusive. By identifying androgen-responsive TFs with binding motifs near the *AS2* TSS, future studies could further characterize the role of this lncRNA in PCa.

## Figures and Tables

**Figure 1 ncrna-08-00081-f001:**
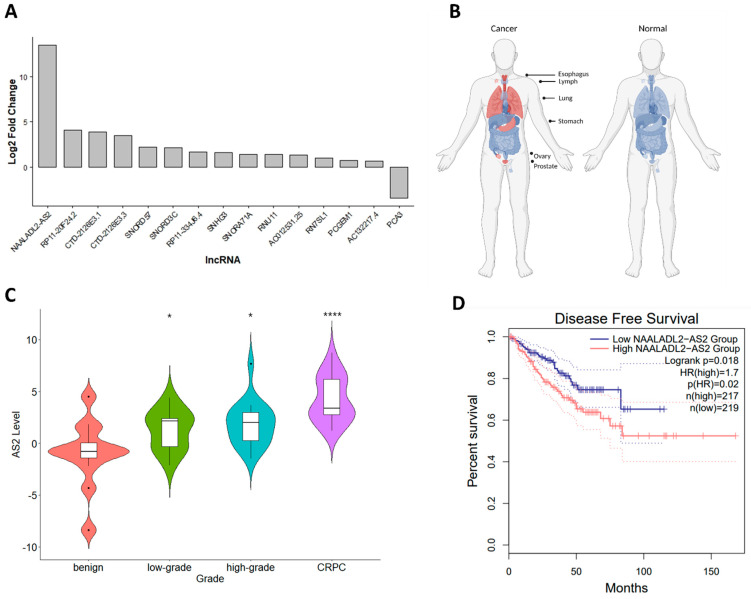
The expression of *AS2* in CRPC versus HSPC (**A**). Graphical representation of *AS2* expression (red) in various malignant (left, TCGA data) and non-malignant (right, GTEx data) accessed via GEPIA2 [12], stylized using BioRender.com (**B**). *AS2* expression as determined by qPCR in benign, hormone sensitive low-grade PCa (Gleason score < 7), hormone sensitive high-grade PCa (Gleason score ≥ 7), and CRPC. Pairwise means comparison against the benign group was done using a *T*-test, *: *p* < 0.05, ****: *p* < 0.0001. (**C**). Kaplan Meier survival analysis of HSPC patients with high and low tumor *AS2* expression (median cut off) (PRAD TCGA data) [12] (**D**).

**Figure 2 ncrna-08-00081-f002:**
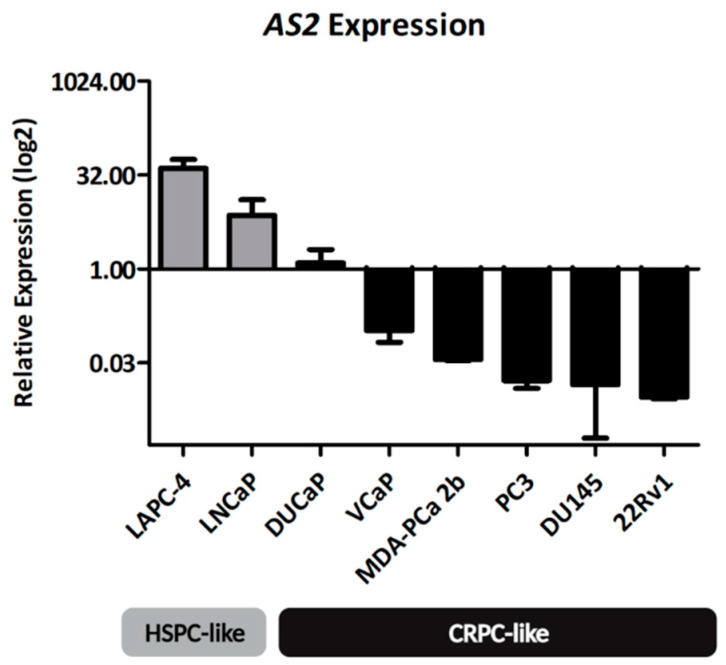
The expression of *AS2* is high in HSPC-like cell lines, LAPC-4 and LNCaP, when compared to CRPC-like PCa cell lines.

**Figure 3 ncrna-08-00081-f003:**
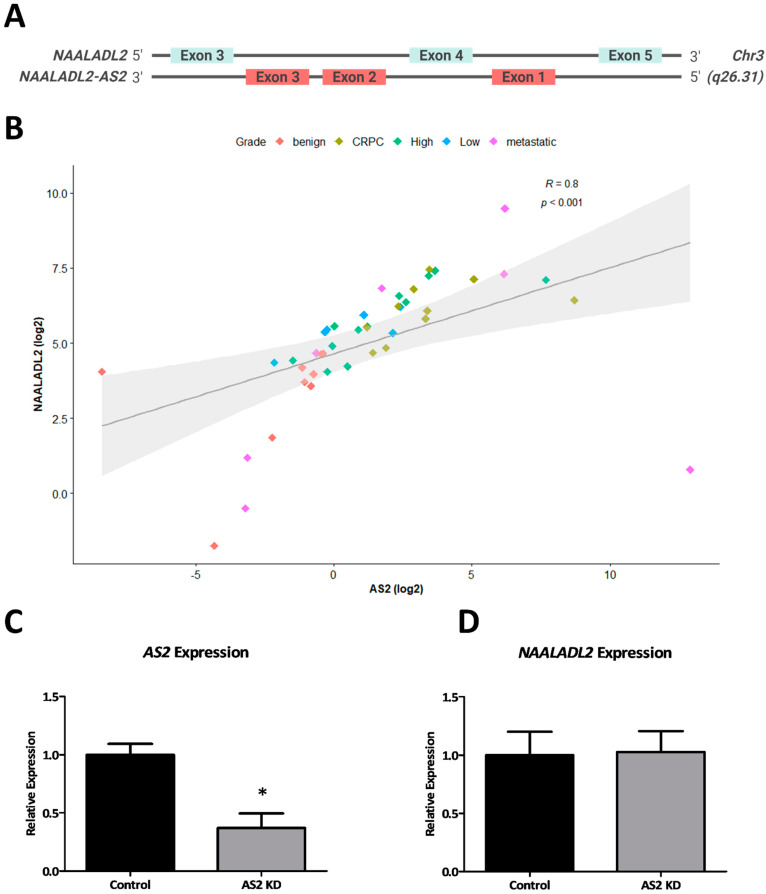
The genomic structures of *NAALADL2* and *AS2* as interpreted from the UCSC Genome Browser (GRCh38/hg38) (not to scale) (**A**). The expression of *AS2* and *NAALADL2* in tumor RNA samples measured by qPCR (*n* = 44), Spearman correlation (**B**). *AS2* and *NAALADL2* expression measured by qPCR in LAPC-4 cells cultured with scrambled (control) or AS2-specific GapmeR antisense oligonucleotides (AS2 KD) for 72 h; Pairwise means comparison was done using a *T*-test, *: *p* < 0.05 (*n* = 3) (**C**,**D**).

**Figure 4 ncrna-08-00081-f004:**
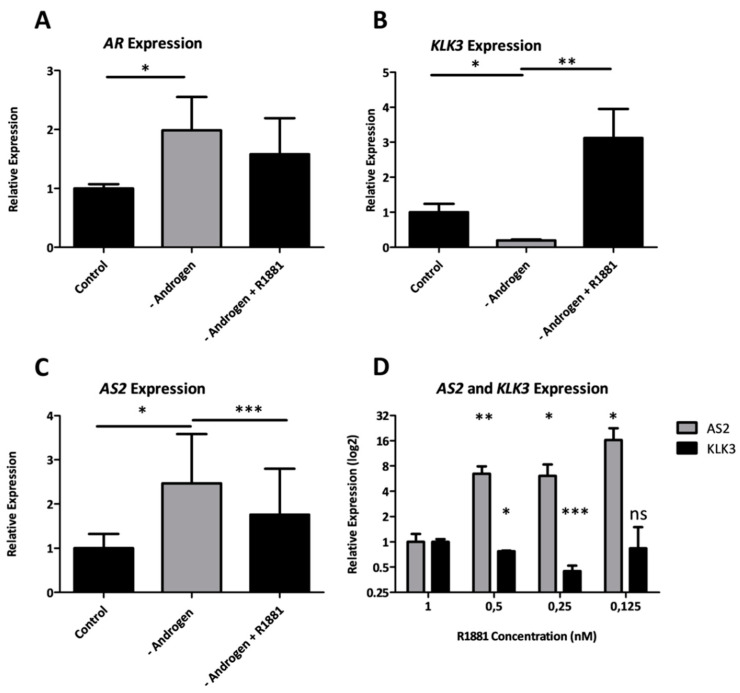
The expression of *KLK3*, *AR*, and *AS2* measured via qPCR in LNCaP cells cultured with fetal bovine serum (Control), charcoal stripped serum (CSS) (-Androgen) or CSS with synthetic androgen R1881 (-Androgen + R1881) (**A**–**C**). *AS2* and *KLK3* expression measured via qPCR in LAPC-4 cells cultured in decreasing concentrations of R1881 (**D**). Pairwise means comparison was done using a *T*-test, *: *p* < 0.05, **: *p* < 0.01, ***: *p* < 0.001.

**Figure 5 ncrna-08-00081-f005:**
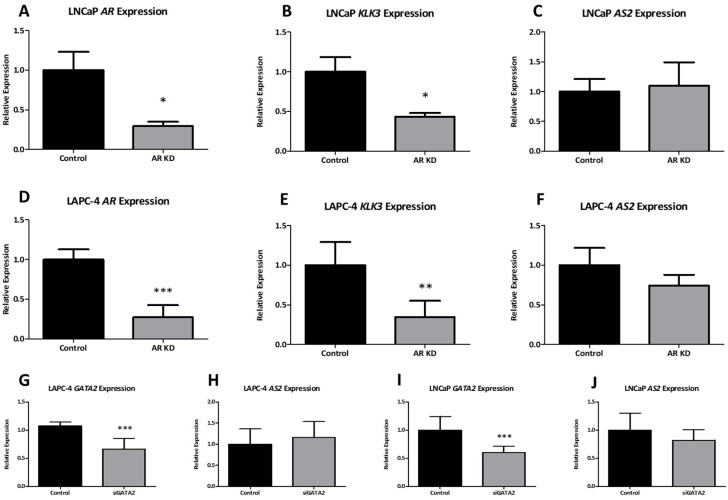
The expression of *AR*, *KLK3,* and *AS2* was measured via qPCR in LNCaP and LAPC-4 cells following exposure to siRNAs targeting *AR* and *GATA2* mRNA. All experiments were performed at *n* ≥ 3. Pairwise means comparison was done using a *T*-test, *: *p* < 0.05, **: *p* < 0.01, ***: *p* < 0.001.

**Figure 6 ncrna-08-00081-f006:**
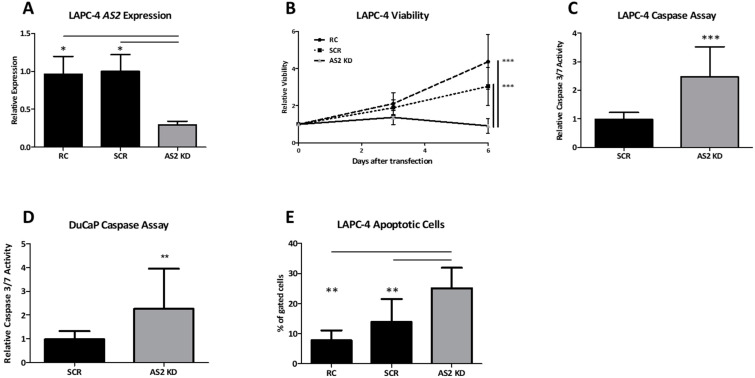
*AS2* expression was measured in LAPC-4 cells after 72 h of exposure to transfection reagent (RC), scrambled GapmeR (SCR), and *AS2*-specific GapmeRs (AS2 KD) (**A**). Viability was assessed on day 0, 3, and 6 days after the transfection of LAPC-4 cells (**B**). Caspase activity was assessed 2 days after transfection in both LAPC-4 and DuCaP cells (**C**,**D**). DNA fragmentation was measured via flow cytometry 5 days following transfection in LAPC-4 cells (**E**) All experiments were performed at *n* ≥ 3. Pairwise means comparison was done using a *T*-test, *: *p* < 0.05, **: *p* < 0.01, ***: *p* < 0.001.

**Figure 7 ncrna-08-00081-f007:**
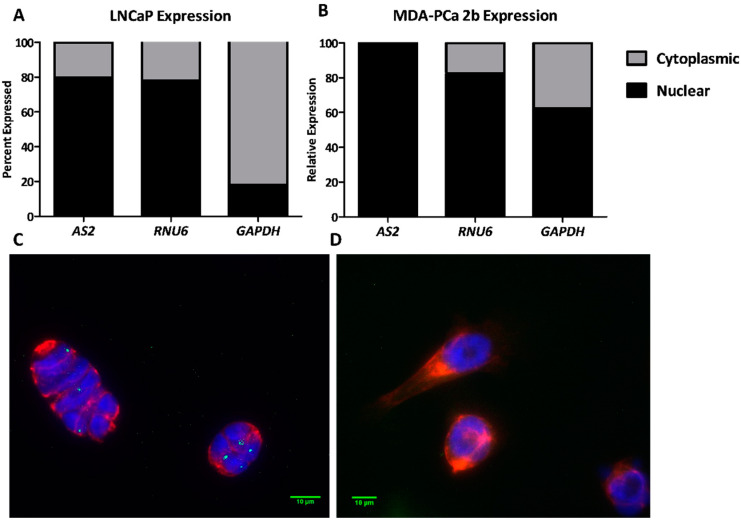
*AS2, RNU6*, and *GAPDH* transcripts levels were analyzed via qPCR in the cytoplasmic and nuclear fractions of LNCaP and MDA-PCa 2b cells (**A**,**B**). *AS2* RNA transcripts (green), cytokeratin protein (red), and nuclear DNA (blue) were visualized using RNA in situ hybridization and immunofluorescence in LAPC-4 (**C**) and PC3 (**D**), respectively.

**Figure 8 ncrna-08-00081-f008:**
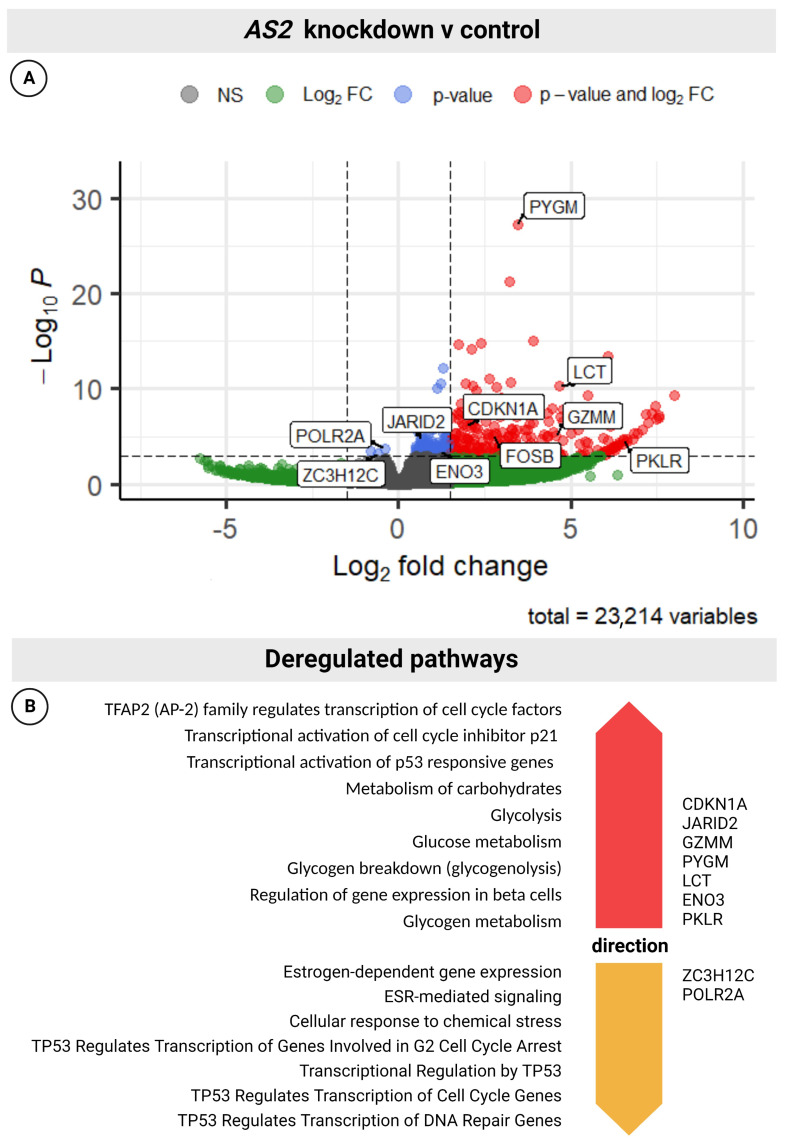
Volcano plot showing the DEGs between *AS2* knockdown (AS2 KD) and reagent control (**A**). Pathways which are deregulated upon *AS2* knockdown, up regulated pathways shown in orange, down regulated pathways shown in yellow (**B**). Statistical significance for the selection of DEGs was set to padj < 0.001, *n* = 3. Figure was slylized using BioRender.com.

## Data Availability

The datasets generated in this study are available upon request to the corresponding author.

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
