# Peer review of "The Androgen Regulated lncRNA NAALADL2-AS2 Promotes Tumor Cell Survival in Prostate Cancer"

_ncrna, 2022, doi:10.3390/ncrna8060081_

Round 1

Reviewer 1 Report

Minor spell/ grammar checks

Conclusions section can be improved by addition of future perspectives 

Author Response

Dear reviewer,

Thank you for your interest towards our work. I have reviewed the manuscript, with a focus on the introduction and conclusion. Following the needed adjustments and changes I believe we have improved the manuscript. As suggested, I have added our thoughts with respects to future perspectives to the conclusion section. 

Sincerely,

Levi Groen

Reviewer 2 Report

The reported results are quite interesting. Some suggestions may be taken to improve the quality of this study.

1. Your experimental results showed different results for tumor-derived RNA and cell lines. In tumor-derived RNA, AS2 was high in CRPC, whereas in cell lines AS2 was high in HSPC. How to explain this phenomenon? It would be better to explain this in part Discussion.

- Tumor-derived RNA : AS2 was found to be significantly upregulated in CRPC when compared to normal, benign hyperplasia, low grade, and high grade HSPC (Figure 1C).

- Cell lines : The expression of AS2 was found to be relatively high in LAPC-4 and LNCaP cells (HSPC-like), and low in the other cell lines (CRPC-like), suggesting that AS2 may be  primarily important for HSPC cells (Figure 2).

2. Page 6, lines 143-144. Picture description and picture order do not match. Correct the order of the pictures in the order of description.

Author Response

Dear reviewer,

Thank you for your interest in our work. Your comments were well received and I believe they have improved our manuscript. I have rearranged the order of figure 4 to match the results described. Furthermore, I believe the discordance between AS2 expression levels in CRPC tissue RNA and cell lines to be due to their relative exposure to anti-androgen therapy. Where CRPC-like cell lines are continuously cultured under "ADT", CRPC-tissue specimens are relatively naïve to successive lines of ADT and remain largely dependent on canonical AR signaling. As shown in our HSPC-like in vitro models, exposure to first-line ADT increases their expression of AS2, potentially as a survival adaptation. In CRPC-like cell lines, non-canonical AR signaling and other adaptations likely make increased AS2 expression redundant. I have included this in the discussion section as suggested.

Thank you,

Levi Groen